# Analysis of the impact of RCEP on the industrial and innovation chains of China's textile and clothing industry

**Li Yang[1], Pivithuru Kumarasinghe[2]***

1 Director of Cross Border E-Commerce Teaching and Research Office, School of Economics and Social Welfare, Zhejiang Shuren University, Hangzhou, China, 2 School of Economics and Social Welfare, Zhejiang Shuren University, Hangzhou, China

* janak_kumarasighe@zjsru.edu.cn

**Data Availability Statement:** All relevant data for this study are publicly available with the title "RCEP on the Industrial and Innovation Chains of China's Textile and Clothing Industry" from the Mendeley

## Abstract

This research examines the impact of the Regional Comprehensive Economic Partnership (RCEP) on the textile and apparel industry within its member nations. The study seeks to understand the implications of RCEP on trade dynamics, innovation chains, and industrial integration in the textile sector. The study uses both quantitative analysis of trade data and qualitative assessment of policy frameworks to analyze changes in textile trade and patterns among RCEP members through UN Comtrade data. Qualitative analysis is conducted to examine RCEP policies related to intellectual property protection, investment regulations, and innovation cooperation. The findings reveal a significant increase in textile trade volume among RCEP member countries following the agreement's implementation. China emerges as a key player, experiencing substantial growth in textile exports to RCEP nations, particularly driven by tariff reduction initiatives. RCEP provisions stimulate demand for innovation within the textile industry, fostering collaborative efforts in scientific research and development.

## 1. Introduction

After eight years of negotiations, the Regional Comprehensive Economic Partnership (RCEP) was signed in November 2020. It officially took effect on January 1, 2022, and became fully effective for all 15 signatory countries on June 2, 2023. This marked a new full implementation stage for the free trade area with the world's largest population, economy, and trade scale. The RCEP has played a significant role in improving the trade and investment levels in the entire region. It has effectively promoted the establishment of a more efficient, closer, mutually beneficial cooperation system among its members for industrial and supply chains [1,2]. This has added continuous momentum to the region's sustained economic recovery and development. Chinese President Xi Jinping has also advocated the "deployment of innovation chains around industrial chains and the layout of industrial chains around innovation chains" to promote high-quality economic development in China. The study aims to analyze the RCEP's impact on regional trade dynamics, economic integration, and industrial development. Through

Data repository (https://doi.org/10.17632/sgvmg6c9zz.1).

**Funding:** This study received support from the Research project of the Zhejiang Provincial Department of Education (Y202351899) and Zhejiang Federation of Humanities and Social Sciences Circles Base research projects (2014JDZ01).

**Competing interests:** The authors have declared that no competing interests exist.

empirical analysis and strategic insights, we aim to better understand the RCEP's role in facilitating sustainable economic growth and enhancing regional cooperation in the post-pandemic era.

## 2. Literature review

The concept of industrial and innovation chains has long been around in economic theories. Adam Smith's theory of specialization was the starting point for an industrial chain. It is a series of interconnected industries with input-output relationships. An industrial chain spans multiple industries upstream and downstream of the production process, forming a chain-like industrial organization [3]. Innovation chains refer to activities involving multiple stakeholders at different stages, aiming to bring about innovation [4]. Innovation is mutual feedback between internal technological drivers and external market demand pulls in innovation chains, which leads to introducing new products or processes [5]. Michael Porter's theory of competitive advantage highlights the integration of production and innovation factors along the chain according to market demands. This integration enhances production efficiency and innovation capabilities [6].

Since the signing and implementation of the RCEP, scholars have been interested in its economic effects. Reduced tariffs have led to significant trade effects and favorable conditions for multinationals to redesign production based on member countries' comparative advantages [7]. RCEP has encouraged the East Asian region to rely more on internal economic circulation and reduce its dependence on the US market. Participation in international economic activities influences innovative activities [8]. Regional trade agreements can enhance government credibility, strengthen mutual credibility between member countries, and incentivize the formation of innovative cooperation relationships within the international innovation cooperation network [9]. These agreements provide channels for introducing cutting-edge technologies and knowledge spillovers, improving technological innovation in developing economies [10].

The textile industry's industrial and innovation chains have been undergoing rapid evolution. They are integrating new technologies, embracing workplace innovations, adopting sustainable efficiencies, and inventing products and processes to meet the changing demands of global consumers and markets [11]. This evolution is driven by the need for increased efficiency and productivity and the desire to create more sustainable and environmentally friendly practices. Innovation in the textile industry involves creating new products and improving processes and business models. For instance, investments in logistics, telecommunications, predictive technologies, and manufacturing processes have helped countries integrate into more lucrative value chains and improve the lives of millions of people [12,13]. China is a significant player in the global textile and apparel industry and is a member of the RCEP [14]. As a result, the country has undergone significant changes in its industrial and innovation chains.

The RCEP agreement has played a pivotal role in driving the transformation and upgrading of China's textile industry, leading to the integration of its industrial and innovation chains. The impact of the RCEP on China's textile industry can be examined from two perspectives: Firstly, from an industrial chain perspective, the RCEP agreement has facilitated the free flow of goods among member states, which has helped reduce trade barriers and improved the efficiency of China's textile industrial chain [14]. This has resulted in China's better integration into the regional industrial chain and enhanced the competitiveness of its textile industry. Secondly, from an innovation chain perspective, the RCEP agreement has encouraged technological exchanges and cooperation among member states. This has allowed China's textile industry to access advanced technologies and innovative ideas, promoting the development of its innovation chain.

Sustainability has become a critical aspect of the textile industry. With the increasing demand for eco-friendly products and more stringent regulations, business players and policy-makers need to develop sustainability innovation in the textile industry. This includes practices like eco-design, eco-label, life cycle assessment, cleaner production, ecoefficiency, waste handling, supply chain management, and enzymatic textile processing. This article analyzes the impact of RCEP on China's textile industrial chain and innovation chain based on authoritative data sources and the RCEP agreement text.

## 3. Methodology

The analysis of the impact of the RCEP on the industrial and innovation chains of China's textile and clothing industry employs a descriptive research methodology. This methodology was chosen because it is suitable for describing the industry's characteristics, situation, and trends without manipulating variables, which aligns with the objectives of this study.

The descriptive research design was deemed appropriate as it allows for the systematic collection and analysis of data to describe the industry's current state and objectively assess RCEP's impact. Given that the researcher has no control over the variables in the study, a descriptive research design provides a structured framework for examining the complex interactions and dynamics within the textile and clothing industry post-RCEP implementation.

Data for this study were collected from various sources, including the UN Comtrade database, data from the Ministry of Commerce of China, and the text of the RCEP agreement. All data underlying the findings of this study are freely available to other researchers. The data for this research, titled "RCEP on the Industrial and Innovation Chains of China's Textile and Clothing Industry," is accessible through Mendeley Data [15]. These sources offer a diverse range of information, encompassing trade patterns, investment flows, and regulatory frameworks pertinent to the textile and clothing industry within the context of RCEP. The collected data were analyzed using tables and ratios to identify patterns, trends, and relationships within the textile and clothing industry post-RCEP. Tables presented quantitative data in a structured format, facilitating comparisons and interpretations. At the same time, ratios were calculated to assess key performance indicators and measure the impact of RCEP on industrial and innovation chains.

In order to ensure that the research findings are reliable and valid, we implemented several measures. Firstly, we collected data from authoritative sources known for accuracy and credibility, such as the UN Comtrade database and official government sources. Secondly, we designed the research methodology to be systematic and replicable, enhancing the study outcomes' reliability. Additionally, we employed triangulation of data from multiple sources to corroborate findings and minimize bias, thereby enhancing the validity of the research. Finally, we conducted the research process transparently, with clear documentation of data collection and analysis procedures, enabling scrutiny and verification by peers and stakeholders. These measures collectively contribute to the robustness and trustworthiness of the research findings.

## 4. Results and findings

The subsequent section delves into the results and findings obtained from the analysis of RCEP's impact on the textile industry. Through comprehensive examination and evaluation of various factors such as trade dynamics, innovation integration, and industrial transformation, this section aims to shed light on the specific outcomes and implications of RCEP implementation for the textile sector within the region.

## 4.1 Trade dynamics of textile and apparel products within RCEP member nations

China has always been a major player in the global textile industry. A report by CCID Consulting titled "2022 China's Top 100 County Economies Research" reveals that 8 out of the top 10 counties in the ranking have the textile industry as their main industry. In fact, in 43 countries with a GDP exceeding one trillion yuan, the textile and clothing industry is considered a crucial pillar for economic development. This has led to a clustering and scaling effect in the industry. Since 2000, the textile and clothing trade scale in RCEP countries has expanded. From 2000 to 2021, the world's exports of textile and clothing trade increased by 132%, while the exports of RCEP countries increased by 290.7%, more than twice the global growth rate. Among them, China's exports have grown by 517.2%, nearly four times the global growth rate. From 2000 to 2021, the share of RCEP textile and clothing exports worldwide increased from 28.73% to 48.38%, while China's share has increased from 13.54% to 36.03%. As a result, the RCEP region has become the world's most important textile and clothing manufacturing center. Fig 1 shows the relationship between the RCEP and China's worldwide textile and clothing industry export share.

In 2021, out of the top 20 countries and regions that exported textiles and clothing globally, three were three members of the RCEP: China, Vietnam, and Japan. These three countries together accounted for 54% of the total exports of the top 20 countries and regions, with China alone contributing 44.9% of the total share. In 2022, China continued to maintain its leading position in the exports of textiles and clothing, with a year-on-year growth rate of 4.9%. Table 1 lists the top 20 countries that export textiles and garments.

On the demand side, countries such as Japan, South Korea, and Australia are significant import markets for textile and clothing products worldwide. In 2022, their imports of clothing products (HS61 and 62) reached $25.1 billion, $12.2 billion, $9.6 billion, and $8.7 billion, respectively. China's vast consumer market is experiencing an increasing demand for imports, while the Association of Southeast Asian Nations (ASEAN), as a rapidly developing garment manufacturing base, also has a massive import scale for textile intermediate products. Therefore, from the global textile and clothing trade landscape perspective, the RCEP region is not

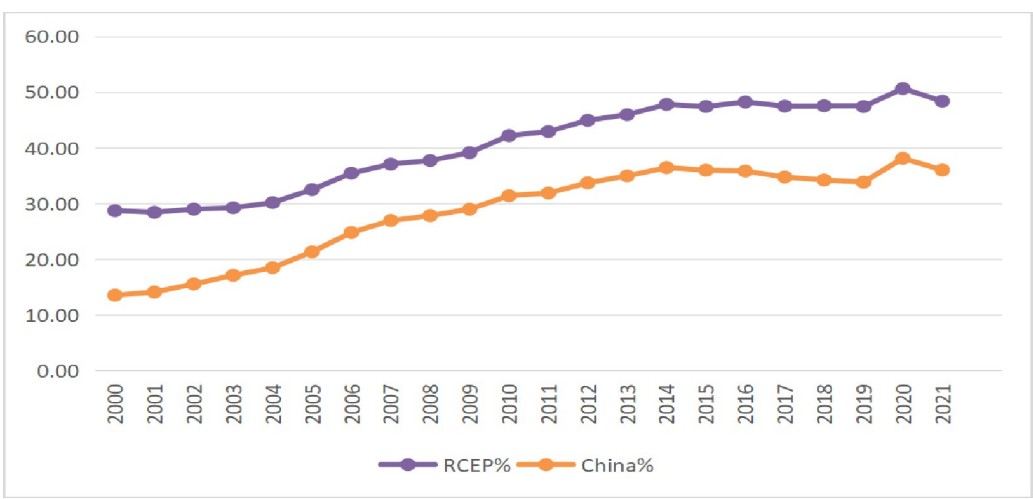

Note: Date from UN Comtrade

**Fig 1. RCEP and China textile and clothing industry export share in the global market.**

**Table 1. Top 20 export countries of textiles and garment.**

| Country | 2021 (Billions of dollars) | proportion (%) | 2022 (Billions of dollars) | proportion (%) |
|---|---|---|---|---|
| China | 3047.46 | 44.90 | 3196.96 | 45.41 |
| Germany | 421.05 | 6.20 | 411.97 | 5.85 |
| India | 414.68 | 6.11 | 383.07 | 5.44 |
| Viet Nam | 412.91 | 6.08 | 454.23 | 6.45 |
| Italy | 376.68 | 5.55 | 423.72 | 6.02 |
| Turkiye | 344.55 | 5.08 | 352.63 | 5.01 |
| USA | 255.43 | 3.76 | 308.13 | 4.38 |
| Netherlands | 210.72 | 3.10 | 206.02 | 2.93 |
| Spain | 206.75 | 3.05 | 208.94 | 2.97 |
| France | 184.27 | 2.71 | 198.63 | 2.82 |
| Poland | 149.78 | 2.21 | 140.07 | 1.99 |
| China, Hong Kong SAR | 134.74 | 1.98 | 97.80 | 1.39 |
| Rep. of Korea | 122.21 | 1.80 | 117.45 | 1.67 |
| United Kingdom | 87.58 | 1.29 | 81.08 | 1.15 |
| Mexico | 78.56 | 1.16 | 91.01 | 1.29 |
| Belgium | 74.04 | 1.09 | 95.57 | 1.36 |
| Japan | 73.75 | 1.09 | 71.48 | 1.02 |
| Denmark | 68.00 | 1.00 | 72.38 | 1.03 |
| Portugal | 64.02 | 0.94 | 64.15 | 0.91 |
| Sri Lanka | 60.70 | 0.89 | 64.48 | 0.92 |
| total | 6787.88 | 100.00 | | 100.00 |

Note: Textile and clothing products under HS code 50–63 (UN Comtrade) are included. Vietnam's 2022 data have yet to be reported.

only the most important textile and clothing manufacturing center in the world but also a rapidly growing and significant consumer market with the most extraordinary global growth potential.

## 4.2 Textile trade dynamics between China and RCEP member nations

Table 2 below provides detailed information on the trade volume and proportion of China's textile and clothing exports to the other 14 member countries of RCEP for 2022. China's exports of textile and clothing products to RCEP member countries have shown growth, and in 2022, the total export value reached US$95.02 billion, representing a 9.3% year-on-year increase. Laos was the export market with the highest growth rate for China's textile exports.

From the perspective of textile and clothing product classification, in 2022, the export growth rates of yarn, textile fabrics, textile products, and clothing between China and RCEP countries were superior to those of similar products traded with the rest of the world. China's imports of textile products decreased during this year, but the decline with RCEP countries was also smaller than that of similar products traded with the rest of the world. Fig 2 compares the growth rates of China's textile product exports to global and RCEP countries in 2022.

For textile machinery products closely associated with textile and clothing production, only HS8447 (knitting machines and stitch-bonding machines) exhibited a higher export growth rate from China to RCEP countries compared to similar products exported globally. Conversely, for HS8445 (machinery for pre-treatment of textile fibers and textile production), China experienced a higher import growth rate from RCEP countries than from the rest of the world for similar products. Fig 3 compares China's export growth rate of textile machinery products to global and RCEP countries in 2022.

**Table 2. China's textile product exports to RCEP member countries in 2022.**

| Country | Export (billions of dollars) | on-year export growth (%) | The proportion of exports in RCEP (%) |
|---|---|---|---|
| ASIAN | 561.38 | 14.17 | 59.55 |
| Brunei Darussalam | 0.26 | -19.99 | 0.03 |
| Myanmar | 34.53 | 44.95 | 3.66 |
| Cambodia | 47.04 | 10.55 | 4.99 |
| Indonesia | 65.02 | 10.91 | 6.90 |
| Lao People's Dem. Rep. | 1.48 | 180.04 | 0.16 |
| Malaysia | 67.39 | 4.54 | 7.15 |
| Philippines | 65.52 | 4.44 | 6.95 |
| Singapore | 34.69 | 82.09 | 3.68 |
| Viet Nam | 192.07 | 8.22 | 20.37 |
| Thailand | 53.39 | 26.75 | 5.66 |
| Australia | 76.62 | 14.66 | 8.13 |
| Japan | 195.57 | 1.43 | 20.74 |
| Rep. of Korea | 99.61 | 3.46 | 10.57 |
| New Zealand | 9.58 | -2.04 | 1.02 |
| Total | 942.76 | 9.96 | 100 |

Note: Date from UN Comtrade.

## 4. 3. Impact of RCEP implementation on the reconstruction and structure of the textile industry supply chain

The textile industry chain involves various components, including raw materials like cotton, silk, wool, polyester, and chemical fibers, the loom industry, weaving and dyeing, apparel, home textiles, and light textile markets [16]. The textile and clothing industry is essential for technological, design, and business model innovation, integrating manufacturing and service economies. With the rise of economic globalization, the international division of labor has shifted towards intra-product trade, and industrial chains have transcended national boundaries, giving rise to global industrial chains.

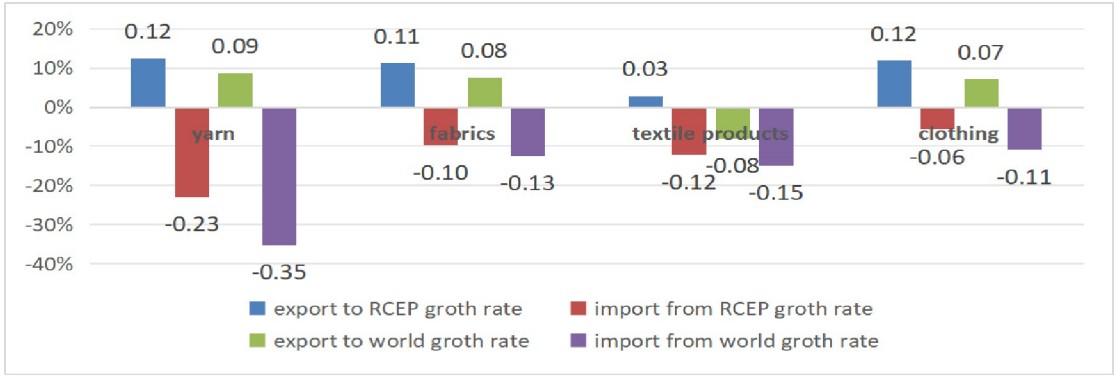

Note： Data from UN Comtrade

**Fig 2. Comparative growth rate of China textile product exports to global and RCEP countries in 2022.**

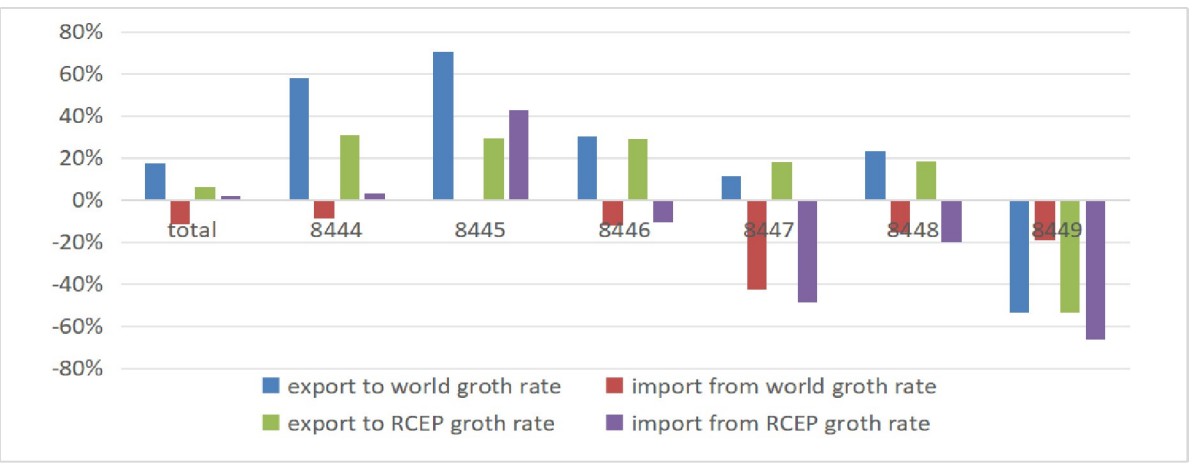

Note: Date from UN Comtrade

**Fig 3. Comparative export growth rate of China textile machinery products to global and RCEP countries in 2022.**

Within the RCEP countries, there are differences in the competitive industries of each country within the textile and clothing industrial chain. China has advantages in areas such as synthetic filament yarn, synthetic staple fibers, nonwovens, unique woven fabrics, industrial textiles, knitted fabrics, knitted clothing, and made-up textile articles. Due to its historical and technological accumulation, Japan has solid technological advantages in high-end fibers such as carbon fiber. Korea has technological solid advantages in fibers such as spandex, and both Japan and Korea have certain advantages in nonwoven, knitted clothing and made-up textile articles. Australia and New Zealand have vast pastures and high-quality wool, thus having advantages in cotton and wool. ASEAN countries have shown advantages in the clothing sector and some textile areas.

Several ASEAN countries have identified the textile and clothing industry as critical in promoting economic development. For example, Vietnam, Cambodia, and Indonesia have developed significant textile and clothing sectors. These governments have introduced many supportive policies in areas such as investment, finance, approval, and tariff concessions to facilitate this. These policies have attracted textile and clothing companies worldwide to invest and build factories in these countries [2]. These countries have been known to introduce various supportive policies to stimulate economic development and attract investment. Vietnam's main advantages are finished textile products (spinning) and carpets. Indonesia is highly competitive in the carpet sector, Thailand is highly competitive in synthetic staple fibers, and Singapore is highly competitive in synthetic filament yarn.

China strategically aligns with RCEP member countries in the textile industry chain. ASEAN and Japan are key partners, ranked as the third and fourth largest export markets for China's textile industry. Korea and Australia, as globally significant textile and clothing consumption markets, are crucial export destinations for China's textile industry end products. In terms of imports, ASEAN has emerged as China's primary source of imported textile and clothing products. Japan and Korea are vital sources of imported functional fabrics, chemical fiber textile clothing, textile dyes, and other products for China. Australia and New Zealand also significantly provide China with a wealth of high-quality wool and other raw textile materials.

## 4.4. The impact of RCEP on the textile industrial chain within the region

The impact of the RCEP on the textile industrial chain within the region is multifaceted. One significant aspect is the tariff reduction under the RCEP, which has led to notable changes in trade dynamics and industrial competitiveness.

**4.4.1. Tariff reduction under the RCEP.** The RCEP aims to remove tariff and non-tariff barriers to trade within the region. This is expected to reduce trade costs and product prices significantly. Over 90% of intra-regional trade in goods will eventually be tariff-free due to the RCEP. China, Japan, and South Korea have established their first free trade agreements, which marks a significant step in their economic and trade relations. This will reduce trade costs between China and Japan, with the proportion of Chinese products with zero tariffs on Japan eventually reaching 86% and the proportion of Japanese products with zero tariffs on China reaching 88%. Upon implementing the RCEP, Japan immediately reduced tariffs to zero for 33.7% of Chinese textile and clothing products, with zero-tariff rates reaching 71.3%, 99.3%, and 99.3% within 11 years, 16 years, and 21 years, respectively. Meanwhile, China immediately reduced tariffs to zero for 10% of Japanese textile and clothing products, with zero-tariff rates reaching 83.3%, 90.7%, and 91.8% within 11 years, 16 years, and 21 years, respectively.

However, Japan's average annual tariff reduction on Chinese textile and clothing products is only 0.5% to 0.8%, which limits the promotional effect on Chinese exports to Japan in the short term. The more significant the proportion of Chinese products with tariffs gradually reduced to zero, the less significant or even damaging the growth of China's exports of such products to Japan after the implementation of the RCEP. Table 3 shows the tariff reduction and export growth of textiles exported by China to Japan.

**4.4.2. Rules of Origin under the RCEP.** The rules of origin for textile and clothing products within the RCEP are relatively lenient. For most products under chapters 50–56 and all products under chapters 57–63, the criterion for determination is "chapter change" without additional conditions. This means that if all non-originating materials used in producing goods have changed at the first two digits of the Harmonized System (HS), they are considered originating.

**Table 3. Tariff reduction and export growth of textiles exported by China to Japan.**

| HS code | Goods Quantity with 10–15 Year Annual Tax Reduction (A) | Total Goods Quantity (B) | Proportion (A/B) | Export growth rate in 2022 |
|---|---|---|---|---|
| 50 silk | 17 | 41 | 41.46% | 20.97% |
| 51wool | 19 | 69 | 27.54% | 51.13% |
| 52cotton | 312 | 385 | 81.04% | -1.67% |
| 53vegetable textile fibres | 6 | 28 | 21.43% | 9.40% |
| 54filaments | 82 | 226 | 36.28% | 7.16% |
| 55staple fibres | 98 | 277 | 35.38% | 12.91% |
| 56 wadding, felt, and nonwoven | 25 | 86 | 29.07% | 0.13% |
| 57 carpets, etc. | 29 | 34 | 85.29% | -3.75% |
| 58 Specially woven fabrics | 24 | 95 | 25.26% | -10.43% |
| 59 Industrial textiles | 25 | 37 | 67.57% | -2.07% |
| 60Fabrics, knitted or crocheted | 52 | 100 | 52.00% | -1.21% |
| 61 knitted or crocheted apparel | 246 | 278 | 88.49% | 0.77% |
| 62 not knitted or crocheted apparel | 170 | 231 | 73.59% | 3.36% |
| 63other textiles | 80 | 93 | 86.02% | -1.42% |

Note: The author calculated based on the schedule of Tariff Commitments of Japan and Trade data from UN Comtrade.

Previously, the ASEAN-Japan Free Trade Agreement imposed specific conditions on clothing product exports to Japan. Knitted fabrics must be manufactured within the free trade area to qualify for duty-free treatment in Japan. If Chinese enterprises invested in clothing manufacturing in ASEAN countries and wanted to export to Japan for duty-free treatment, they had to invest in fabric production within the region. Only then would they be considered as originating from ASEAN and qualify for duty-free treatment in Japan (except for least developed countries such as Cambodia and Myanmar).

However, the implementation of the RCEP changed the game. Fabrics imported from China by ASEAN countries are processed into clothing within the ASEAN change HS code chapter. When these products are exported to Japan, they are considered as originating within the RCEP region according to the rules of origin under the RCEP, thus qualifying for duty-free treatment. This means that products that were previously subject to the rules of origin under the ASEAN-Japan and Vietnam-Japan Free Trade Agreements that necessitated local production in ASEAN or were ineligible for duty-free treatment due to the inability to produce in ASEAN can now enjoy duty-free treatment in Japan.

Vietnam and Indonesia are China's largest export markets for upstream textile products, such as yarn and intermediate textile fabrics. In 2022, China exported yarn worth US1.396 billion to Vietnam, an increase of 9.73%. Yarn exports to Indonesia increased by 26.66%, while textile fabric exports increased by 22.37%. During the same period, China's yarn exports to Cambodia, the Philippines, and Myanmar increased by 34%, 46%, and 40%, respectively, while textile fabric exports to Cambodia and Myanmar increased by 5.67% and 43.05%, respectively.

Table 4 presents the growth rate of yarn and textile fabrics exported by China to ASEAN countries in 2022.

ASEAN's clothing exports to Japan increased by 20.85% in 2022 after a sharp decline from 2019 to 2021. According to Fig 4, implementing the RCEP has positively impacted China's exports of intermediate products, such as yarn, to the ASEAN countries.

Additionally, the RCEP facilitates ASEAN countries' making the most of China's production advantages in intermediate products, including yarn and fabrics, to enhance their exports to Japan. This will lead to an optimal allocation of resources within the textile industry production chain.

**4.4.3. Investment provisions of the RCEP.** The investment regulations of the RCEP consist of two parts: textual rules and negative lists. The textual rules are primarily described in Chapter 10 (Investment) and two annexes (Customary International Law and Expropriation). Furthermore, there are provisions related to investment in other chapters of the agreement,

**Table 4. The growth rate of yarn and textile fabrics exported by China to ASEAN.**

| Country | yarn | fabrics |
|---|---|---|
| Brunei Darussalam | -70.35% | -14.26% |
| Myanmar | 40.29% | 43.05% |
| Cambodia | 34.54% | 5.67% |
| Indonesia | 26.66% | 22.37% |
| Lao People's Dem. Rep. | 139.27% | 42.02% |
| Malaysia | 2.57% | -2.79% |
| Philippines | 46.06% | -8.64% |
| Singapore | -30.53% | 118.90% |
| Viet Nam | 9.73% | 7.56% |
| Thailand | 21.98% | 21.92% |

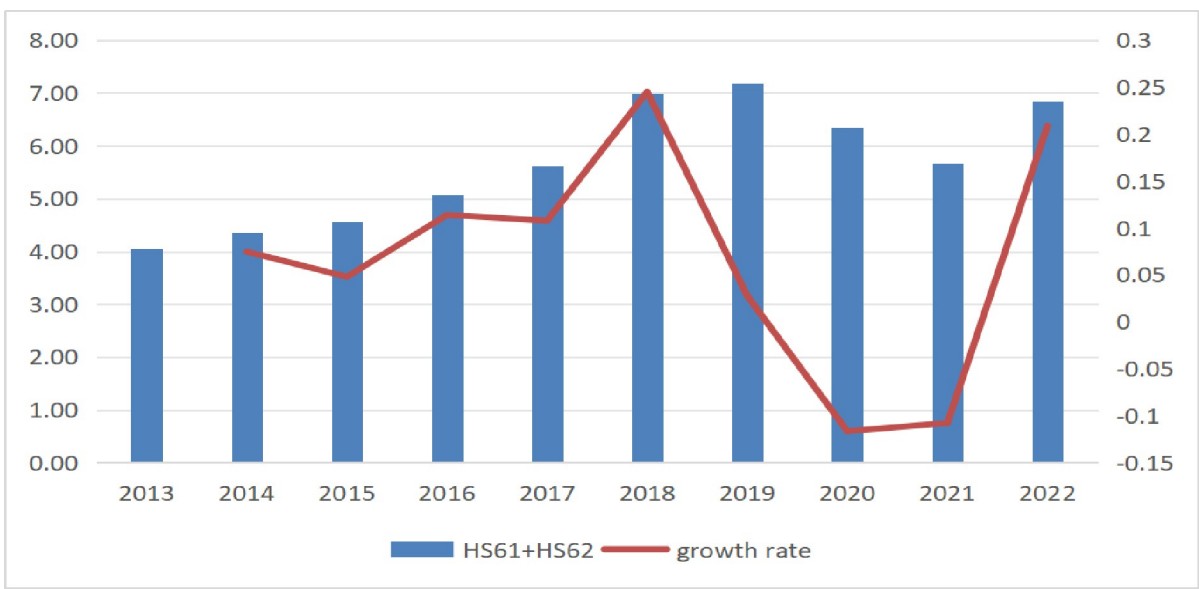

Note: Date from UN Comtrade

**Fig 4. The amount and growth rate of clothing exports from ASEAN to Japan (Billions of Dollars %).**

such as Chapter 1 (Initial Provisions and General Definitions), Chapter 17 (General Provisions and Exceptions), and Chapter 19 (Dispute Settlement). Along with the textual rules, Annex 3 of the RCEP agreement (Schedules of Reservations and Non-Conforming Measures on Services and Investment) lists each member country's negative lists in the investment field [17].

The investment regulations of the RCEP cover four aspects: investment protection, investment liberalization, investment promotion, and investment facilitation. These rules maintain the key content of traditional investment agreements and reflect new advancements in international investment contracting practices. All 15 member countries have made high-level open commitments using negative lists for investments in five non-service sectors: manufacturing, agriculture, forestry, fisheries, and mining. The RCEP's investment rules positively impact attracting foreign investment, creating a favorable business environment, and expanding international cooperation in the textile and apparel industry. Based on data from the Ministry of Commerce of China, China's textile industry directly invested US$3.15 billion in RCEP countries from 2015 to 2020, accounting for 38% of China's overall direct investment during the same period.

Historically, the proportion of mutual investment between China and Japan in their respective total foreign direct investment (FDI) has been relatively low. In 2020 and 2021, Japan's FDI in China accounted for only 2.34% and 2.26% of China's total FDI, respectively. Similarly, China's direct investment in Japan was even less, comprising only 0.32% and 0.43% of China's outbound FDI. This indicates significant potential for growth in bilateral direct investment between China and Japan. Table 5 shows that in 2022, China attracted a 17.68% increase in FDI from Japan, much higher than the overall growth rate of 9.02% for China's FDI and the 6.46% growth rate for FDI from Asia. However, due to factors such as the COVID-19 pandemic, China's outbound investment showed negative growth in 2022. Combined with Japan's economic reasons, China's investment in Japan experienced a steep decline with a growth rate of -47.98%, much lower than the overall growth rate of -8.78%. Table 5 shows Chinese investment in Japan and Japanese FDI in China.

**Table 5. Chinese investment in Japan and Japanese FDI in China (Million USD).**

|      |            | 2020      | 2021      | 2022      | Growth rate |
|------|------------|-----------|-----------|-----------|-------------|
| FDI  | Japan      | 33744.8   | 39132.5   | 46050.8   | 17.68%      |
|      | Total      | 1443692.9 | 1734833.1 | 1891324.1 | 9.02%       |
|      | Proportion | 2.34%     | 2.26%     | 2.43%     |             |
|      | Asia       | 1240254   | 1536446.4 | 1635655.5 | 6.46%       |
|      | Proportion | 2.72%     | 2.55%     | 2.82%     |             |
| OFDI | Japan      | 4868.3    | 7621.4    | 3964.8    | -47.98%     |
|      | Total      | 1537102.6 | 1788193.2 | 1631210   | -8.78%      |
|      | Proportion | 0.32%     | 0.43%     | 0.24%     |             |
|      | Asia       | 1123436.5 | 1281020.5 | 1242835.4 | -2.98%      |
|      | Proportion | 0.43%     | 0.59%     | 0.32%     |             |

Note: Data from the National Bureau of Statistics.

After implementing the RCEP, China achieved a significant breakthrough by using a negative list format to commit to non-service sector investments in a free trade agreement for the first time. Japan has opened up investments in agriculture, forestry, fisheries, and mining sectors, excluding only a few sensitive areas. The RCEP has significantly improved market access between China and Japan and enhanced transparency of their investment policies. This not only benefits Chinese investors in entering the Japanese market and addresses the imbalance in investment between the two countries but also encourages Japanese enterprises to expand their investments in China and strengthen existing investments.

Therefore, in the short term, the implementation of the RCEP has improved the business environment, and its promotional effect on attracting Japanese investment in China is more pronounced.

## 4.5. The impact of RCEP on the layout of the innovation chain in the regional textile industry

The innovation chain in the textile industry refers to a systematic collection of innovative activities that involve various entities and links in the industry. These entities include upstream, midstream, and downstream enterprises [18]. The innovation chain is guided by market demand, and the enterprises integrate innovative resources through various activities. The outcome of the chain is the commercialization of textile and apparel products [19].

The textile innovation chain has a chronological sequence that follows the general innovation chain. It starts with innovative demand, basic research, applied research, design and development, production, and sales, and ends with industrialization and diffusion. Horizontally, the textile innovation chain corresponds to different segments of the industrial chain, such as loom innovation, material innovation, design innovation, manufacturing innovation, supply chain innovation, and marketing innovation [18–20]. This can be more specifically illustrated in the following Fig 5.

RCEP has significantly impacted the regional textile and apparel innovation chain and accelerated advancements in it, enhancing the industry's competitiveness globally.

**4.5.1. Stimulate demand for innovation.** After analyzing the data, it has been found that many countries within the region are significant importers of traditional textile and clothing products globally. Implementing the RCEP leads to an increase in the market size within the region, providing companies with better market opportunities. At the same time, implementing the RCEP has impacted the reconstruction and expansion of the textile industry chain

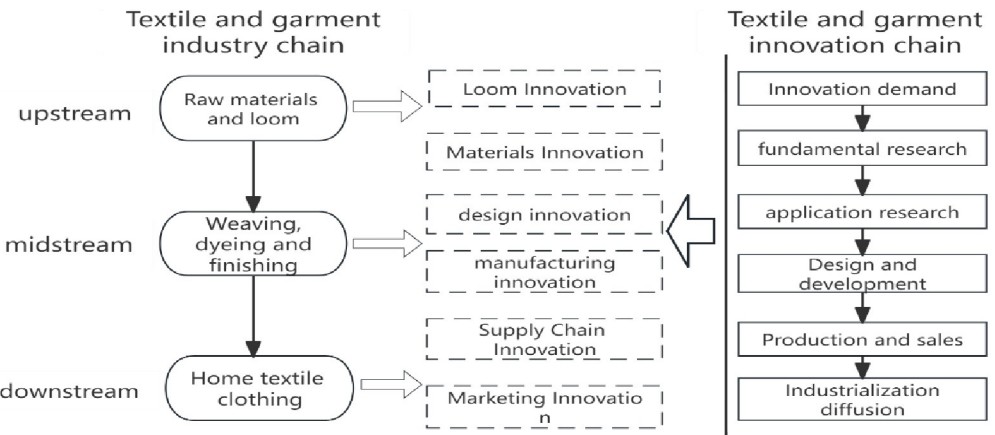

**Fig 5. Textile and garment industry chain and innovation chain.**

within the region. Therefore, based on the vertical composition of the textile innovation chain, the textile industry chain's expansion and the consumer market's growth due to the RCEP will create a higher practical demand for innovative outcomes and stimulate innovation within the region.

**4.5.2. Promote international cooperation in scientific and technological innovation.** International cooperation in technological innovation involves collaboration among countries to acquire knowledge in scientific, technological, and innovative fields. This cooperation can occur in various stages of the innovation chain, including primary and applied research, design, and development [19,20]. The RCEP, a trade agreement between member states, has established a new framework for cooperation in technological innovation within the textile industry.

**4.5.3. Intellectual property protection.** The RCEP, or Regional Comprehensive Economic Partnership, has 83 articles covering various aspects of intellectual property, such as general provisions, basic principles, copyrights, trademarks, geographical indications, patents, industrial designs, and the enforcement of intellectual property rights. With the RCEP provisions, any contracting party can regulate and manage any infringement committed by other contracting parties, thus breaking geographical restrictions. This enables better protection and development of intellectual property within the region.

Moreover, the RCEP agreement's Chapter 14 discusses the importance of small and medium-sized enterprises (SMEs) in enhancing their awareness, understanding, and application of the intellectual property system. This chapter also encourages innovation, improves market access thresholds, and ensures enterprises can engage in technological research and development and transform research outcomes without worrying about technology infringement or plagiarism. Thus, the RCEP framework promotes knowledge sharing and technology transfer within the region, encouraging cooperation among member countries in technological innovation.

**4.5.4. Optimizing the allocation of innovation resources.** The RCEP has established commitment standards that facilitate the movement of natural persons, intellectual property, service trade, and investment. This helps innovation resources flow easily within the region, making the entire region more attractive to external innovation resources. Enterprises can easily access innovation resources from other countries in the region, including technology, talent, and capital, thanks to the RCEP framework. This also contributes to the development of the regional innovation network, enhancing the region's overall innovation capability. The

free movement of talents among RCEP member countries facilitates innovative exchanges and collaborative research, enabling member countries to share scientific and technological resources and research outcomes, thereby promoting the rapid development of technological innovation in the textile and clothing industry.

**4.5.5. Provision of collaborative innovation platforms.** The RCEP member countries can use the platform to strengthen international cooperation in technological innovation. This can be achieved by establishing fixed mechanisms for scientific and technological cooperation, promoting enterprise participation, and encouraging collaboration in new technologies [21,22]. By doing so, member countries can create more opportunities and platforms for cooperation. This will help them jointly develop new technologies, share resources and experiences, and tackle global challenges.

**4.5.6. Promote innovation transformation and dissemination.** The RCEP has helped to reduce trade barriers between its member countries, making it easier for goods and services to move freely within the region. This has allowed innovative technologies and products to enter other member countries' markets more quickly, facilitating their commercialization and industrialization. Additionally, the RCEP's rules of origin have further reduced resource allocation costs and improved efficiency, creating a favorable business environment for foreign-invested enterprises. Both of these factors contribute to the application and transformation of inventions from innovative countries within the industrial chains of member countries.

The RCEP also provides more cooperation opportunities and resource support for innovative countries, promoting the further development and optimization of their inventions. Some Chinese enterprises are even implementing a "Made in China, Manufactured Globally" strategy by transferring part of their production capacity and technology to other RCEP member countries to utilize local resources better and save costs. Meanwhile, these Chinese enterprises focus on research and development of core technologies and production. The RCEP has incorporated data flow and information storage provisions into its institutional framework, creating a relatively lenient institutional environment for data transmission within the region while ensuring national security and personal privacy. Article 11 of Chapter 12 stipulates that "the Parties shall maintain their current practice of not imposing customs duties on electronic transmissions between the Parties," significantly reducing the transaction costs of data transmission among RCEP member countries and promoting cross-border data flow and aggregation, which is conducive to the dissemination of innovation achievements.

## 4.6. The RCEP integrates the textile and apparel industrial and innovation chains

The industrial and innovation chains are interconnected yet distinct systems, mutually supporting, depending on, integrating, and advancing each other. Their integration is evident in the amalgamation of production and innovation entities, processes, and technological advancements [23]. In summary, RCEP facilitates the integration of textile and apparel industrial and innovation chains by merging production entities, processes, and technologies, thereby promoting industrialization and innovation.

**4.6.1. RCEP's role in integrating production and innovation entities in the textile industry.** The production entities in the textile industry are the enterprises along the industrial chain. In contrast, the innovation entities include enterprises, scientific research institutions, and universities engaged in innovative activities in the textile industry [24]. Firstly, the cooperation mechanisms under the framework of the RCEP agreement provide countries with more platforms for exchange and cooperation, including technical seminars, industry matching meetings, and other activities. These events allow production and innovation entities to

understand each other's needs, share experiences and resources, and facilitate deeper cooperation in technological research and development, product innovation, and market development [24]. Secondly, the RCEP provides a more stable and transparent institutional environment, offering better institutional guarantees for cooperation among enterprises, reducing market uncertainty, and encouraging enterprises to integrate production and innovation actively. Furthermore, it also facilitates the exchange and cooperation among the innovation entities in the textile industry of various member countries in the region, contributing to establishing a cross-regional innovation network in the textile industry.

**4.6.2. RCEP's role in integrating the textile industry's production and innovation processes.** Firstly, the RCEP has facilitated technological innovation in the textile production process. Under the framework of the RCEP, textile enterprises can more easily introduce advanced production equipment and technologies from other countries, which can improve production efficiency and provide strong support for their innovation activities. Secondly, technological innovation in the textile industry has better promoted the upgrading of regional industrial chains. Under the RCEP framework, research and development achievements in the textile field by one country's innovation entities can achieve commercial application of the technology in other member countries. Thirdly, the RCEP has strengthened the market orientation in the textile industry's innovation process. Opening national markets under the RCEP framework is conducive to enterprises more accurately grasping market demands and trends, thereby targeting innovation activities.

**4.6.3. RCEP's role in integrating technological advancement and industrialization in the textile industry.** The RCEP has positively impacted technological advancement in the textile industry. Its implementation has resulted in increased competition within the domestic textile market, forcing companies to improve the quality of their products and add more value to them. Consequently, enterprises have been prompted to invest more in technological research and development, accelerate their transformation and upgrading process, and raise their technological capabilities. The intensity of research and development investment in China's textile industry has increased from 0.46% in 2012 to 1.02% in 2022. As of 2022, textile enterprises above a specific size have spent 53.51 billion yuan on R&D, marking a 3.82% increase.

Furthermore, the RCEP has accelerated the progress of technological industrialization. With technological innovation being the driving force behind new industrial boundaries, the development of material technology and industrial textiles is leading to faster integration of the textile industry with various fields such as the health industry, digital economy, bio-economy, green economy, and aerospace, among others. The RCEP has provided a conducive business and legal environment for technological innovation, which has expedited the process of technological industrialization.

## 4.7. Environmental, social, and governance (ESG) goals and impact of RCEP

The RCEP framework encourages the adoption of advanced technologies and practices that reduce the environmental impact of textile production. Member countries are likely to implement cleaner and more efficient production processes, significantly lowering greenhouse gas emissions, reducing waste, and conserving water resources [25]. For instance, introducing eco-friendly materials and energy-efficient machinery can play a crucial role in minimizing the environmental footprint of the textile industry. RCEP promotes harmonizing environmental regulations and standards across member countries [26]. This can lead to more stringent environmental protections and encourage companies to adhere to higher sustainability

standards, fostering a more environmentally responsible industry. The RCEP agreement includes provisions that can positively impact labor standards and working conditions in the textile industry. By promoting fair trade practices and encouraging adherence to international labor standards, RCEP can help improve the well-being of workers in the textile sector. This includes better wages, safer working conditions, and eliminating exploitative labor practices. Enhanced economic activity and investments in the textile sector due to RCEP can lead to greater community development [14]. Increased employment opportunities and improved infrastructure can uplift local communities, contributing to social stability and growth. The RCEP's emphasis on stable and transparent institutional environments supports better corporate governance in the textile industry. Companies are encouraged to adopt more transparent business practices, improve reporting standards, and enhance accountability, which can build investor confidence and drive sustainable growth. The RCEP framework can foster a culture of corporate social responsibility among textile enterprises. By adhering to international best practices and engaging in ethical business conduct, companies can contribute positively to society and the environment, aligning their operations with broader governance goals.

## 5. Conclusion

The Regional Comprehensive Economic Partnership (RCEP) is a pivotal agreement reshaping the dynamics of the member countries' textile and apparel industries. This research elucidates the agreement's transformative effects on trade, innovation, and industrial integration within the region through an in-depth analysis of its impact on various facets of the industry. The Regional Comprehensive Economic Partnership (RCEP) has transformed the textile industry in member countries. RCEP has increased textile trade among members, with China emerging as a dominant player. Tariff reduction initiatives have increased competitiveness and market access for member states. Eliminating tariff barriers and lenient rules of origin have stimulated demand for innovation, driving technological advancements in the industry. Intellectual property protection under the RCEP framework has fostered a conducive environment for knowledge sharing and technology transfer. RCEP has promoted international cooperation and accelerated technological industrialization within the textile industry. It has positioned member countries to thrive in an increasingly interconnected and competitive global market. The RCEP is a pivotal agreement reshaping the dynamics of the member countries' textile and apparel industries. Furthermore, including ESG considerations highlights the RCEP's comprehensive benefits beyond mere economic gains, emphasizing its role in promoting sustainable, socially responsible, and well-governed practices within the textile sector.

## 6. Limitations

The focus is primarily on quantitative trade dynamics and qualitative policy assessments, with less emphasis on technological advancements and geopolitical dynamics. Time constraints limit the inclusion of recent developments. While the findings offer insights into the textile industry within RCEP nations, generalizing them to other industries or regions may be limited. Time constraints restrict capturing recent developments, and the complexity of interactions within the industry poses challenges in fully interpreting relationships. Despite these limitations, the study provides valuable insights into the impact of RCEP on the textile industry and lays the groundwork for future research.

## Author Contributions

**Conceptualization:** Li Yang.

**Data curation:** Li Yang, Pivithuru Kumarasinghe.

**Formal analysis:** Li Yang, Pivithuru Kumarasinghe.

**Writing – original draft:** Li Yang.

**Writing – review & editing:** Pivithuru Kumarasinghe.

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
