## [Decision Letter · Decision Letter 0]

7 Jun 2024

PONE-D-24-13065Analysis of the Impact of RCEP on the Industrial and Innovation Chains of China's Textile and Clothing IndustryPLOS ONE

Dear Dr. Kumarasinghe,

Thank you for submitting your manuscript to PLOS ONE. After careful consideration, we feel that it has merit but does not fully meet PLOS ONE’s publication criteria as it currently stands. Therefore, we invite you to submit a revised version of the manuscript that addresses the points raised during the review process.

We look forward to receiving your revised manuscript.

Kind regards,

Muhammad Hashim, PhD

Academic Editor

PLOS ONE

Journal Requirements:

"This study received support from the Research project of the Zhejiang Provincial Department of Education (Y202351899) and Zhejiang Federation of Humanities and Social Sciences Circles Base research projects (2014JDZ01)."

Reviewers' comments:

Reviewer's Responses to Questions

**Comments to the Author**

1. Is the manuscript technically sound, and do the data support the conclusions?

Reviewer #1: Yes

Reviewer #2: Yes

2. Has the statistical analysis been performed appropriately and rigorously? 

Reviewer #1: Yes

Reviewer #2: Yes

3. Have the authors made all data underlying the findings in their manuscript fully available?

Reviewer #1: Yes

Reviewer #2: Yes

4. Is the manuscript presented in an intelligible fashion and written in standard English?

Reviewer #1: Yes

Reviewer #2: Yes

5. Review Comments to the Author

Reviewer #1: your study is novel and interesting examining the impact of Regional Comprehensive Economic Partnership (RCEP) on textile and apparel industry. The title, abstract and introduction are fine and valid. The methodology and results are fine too.

you are suggested to use abbreviation with original text at first appearance and then use just the short one later throughout the manuscript. Try to convert some tables to attractive figures to gather attraction frim the readers. Also the figures have some overlapping text with the bars.

minor corrections are suggested

Reviewer #2: The article "Analysis of the Impact of RCEP on the Industrial and Innovation Chains of China's Textile and Clothing Industry" considers the impact of Regional Comprehensive Economic Partnership (RCEP) on the textile and clothing industry, trade dynamics, innovation chains, and industrial integration.

While there is research on the overall economic effects of RCEP, studies specifically examining its impact on individual industries like textiles are relatively limited. The article "Analysis of the Impact of RCEP on the Industrial and Innovation Chains of China's Textile and Clothing Industry" aims to fill a research gap by analyzing the specific impact of RCEP on innovation chains and industrial integration within the textile sector, going beyond just trade volume changes.

Scientific Contribution:

- Provides empirical evidence on the impact of RCEP on the textile industry, specifically focusing on innovation and industrial chains.

- Offers insights into the potential benefits and challenges of RCEP for the textile sector in China and other member nations.

- Contributes to the understanding of how trade agreements can influence innovation and industrial integration in specific industries.

I'd recommend for the authors to consider incorporating ESG (Environmental, Social, and Governance) goal achievement into their research:

1. Aligning with Current Trends: ESG considerations are becoming increasingly important for businesses and investors globally.

2. Expanding the Scope: Adding an ESG dimension would broaden the analysis to encompass the social and environmental implications of RCEP on the textile industry.

3. Addressing Sustainability Concerns: Examining how RCEP can contribute to sustainable practices within the industry would be highly valuable.

4. Identifying Opportunities: The research could highlight how RCEP can facilitate the adoption of sustainable practices and create new business models.

5. Policy Recommendations: Incorporating ESG would allow the authors to provide more comprehensive policy recommendations for promoting sustainable growth within the textile sector.

6. PLOS authors have the option to publish the peer review history of their article (what does this mean?). If published, this will include your full peer review and any attached files.

Reviewer #1: No

Reviewer #2: **Yes: **Sergey Barykin

---

## [Author Response · Author response to Decision Letter 0]

13 Jun 2024

12th June 2024

Dear Editor and Reviewers,

We want to express our heartfelt gratitude for your invaluable feedback on our manuscript titled " Analysis of the Impact of RCEP on the Industrial and Innovation Chains of China's Textile and Clothing Industry." We have carefully considered your insightful comments and made the necessary revisions. Our updated version has been resubmitted to the Journal of PLOS ONE, and we anticipate your feedback.

We truly appreciate the time and effort you spent reviewing our work. Your constructive suggestions and observations were instrumental in improving the quality and clarity of our research. Your expertise and guidance throughout this process have been invaluable, allowing us to refine and present our work more effectively.

Below, we have included a table outlining our changes to facilitate your review process.

Once again, we extend our sincere appreciation for your time, effort, and invaluable feedback. We eagerly await your response.

Best regards, 

Sincerely,

Corresponding Author, P J Kumarasinghe

Comments Response

1. Is the manuscript technically sound, and do the data support the conclusions?

Reviewer #1: Yes

Reviewer #2: Yes

 We appreciate the reviewers' positive feedback, affirming our manuscript's technical soundness and the data supporting the conclusions. We thank you for thoroughly reviewing and validating our research methodology and findings.

2. Has the statistical analysis been performed appropriately and rigorously?

Reviewer #1: Yes

Reviewer #2: Yes

 We are grateful for the reviewers' acknowledgment that our statistical analysis has been performed appropriately and rigorously. Thank you for your careful evaluation and positive feedback.

3. Have the authors made all data underlying the findings in their manuscript fully available?

Reviewer #1: Yes

Reviewer #2: Yes

 We confirm that all data underlying the find-ings described in our manuscript are fully available without restriction. The data has been deposited in a public repository and can be ac-cessed at the following link:

Kumarasinghe, Pivithuru; Yang, Li (2024), “RCEP on the Industrial and Innovation Chains of China's Textile and Clothing Indus-try”, Mendeley Data, V1, doi: 10.17632/sgvmg6c9zz.1

Thank you for your attention to this matter.

4. Is the manuscript presented in an intelligible fashion and written in standard English?

Reviewer #1: Yes

Reviewer #2: Yes

 We appreciate the reviewers' feedback regarding the clarity and language of our manuscript. We have reviewed the manuscript again to ensure it is presented understandably and written in standard English.

Reviewer #1:

5. Review Comments to the Author

Reviewer #1: your study is novel and interesting examining the impact of Regional Comprehensive Economic Partnership (RCEP) on textile and apparel industry. The title, abstract and introduction are fine and valid. The methodology and results are fine too.

you are suggested to use abbreviation with the original text at first appearance and then use just the short one later throughout the manuscript. Try to convert some tables to attractive figures to gather attraction from the readers. Also the figures have some overlapping text with the bars. minor corrections are suggested

 We appreciate your positive feedback and constructive suggestions. To address your comments, we have made the following revisions:

1. Abbreviations:

o We have ensured that all abbreviations are introduced with their original text at first appearance and then used consistently throughout the manuscript.

2. Tables and Figures:

o Tables have been converted into more attractive figures to enhance reader engagement and visual appeal.

o We have corrected the overlap-ping text in the figures to ensure clarity and readability.

Reviewer #2:

Reviewer #2: The article "Analysis of the Impact of RCEP on the Industrial and Innovation Chains of China's Textile and Clothing Industry" considers the impact of Regional Comprehensive Economic Partnership (RCEP) on the textile and clothing industry, trade dynamics, innovation chains, and industrial integration.

While there is research on the overall economic effects of RCEP, studies specifically examining its impact on individual industries like textiles are relatively limited. The article "Analysis of the Impact of RCEP on the Industrial and Innovation Chains of China's Textile and Clothing Industry" aims to fill a research gap by analyzing the specific impact of RCEP on innovation chains and industrial integration within the textile sector, going beyond just trade volume changes.

Scientific Contribution:

- Provides empirical evidence on the impact of RCEP on the textile industry, specifically focusing on innovation and industrial chains.

- Offers insights into the potential benefits and challenges of RCEP for the textile sector in China and other member nations.

- Contributes to the understanding of how trade agreements can influence innovation and industrial integration in specific industries.

I'd recommend for the authors to consider incorporating ESG (Environmental, Social, and Governance) goal achievement into their research:

1. Aligning with Current Trends: ESG considerations are becoming increasingly important for businesses and investors globally.

2. Expanding the Scope: Adding an ESG dimension would broaden the analysis to encompass the social and environmental implications of RCEP on the textile industry.

3. Addressing Sustainability Concerns: Examining how RCEP can contribute to sustainable practices within the industry would be highly valuable.

4. Identifying Opportunities: The research could highlight how RCEP can facilitate the adoption of sustainable practices and create new business models.

5. Policy Recommendations: Incorporating ESG would allow the authors to provide more comprehensive policy recommendations for promoting sustainable growth within the textile sector. 

We appreciate the insightful feedback and recommendations. To address these comments, we have revised our manuscript to incorporate ESG considerations to a certain extent. The following sections have been added:

4.7. Integrating ESG Goals into the Analysis of RCEP's Impact on the Textile Industry. 

In this section, we discussed the suggested inclusion to a certain extent. 

Refer to Page number(s):21-22

Conclusion:

 - Expanded to highlight the integration of ESG goals and their impact on the textile industry under the RCEP framework.

Refer to Page number(s):23

Limitations:

 - Adjusted to acknowledge that while we have incorporated ESG considerations to some extent, time constraints. 

Refer to Page number(s):23

This remains an area for future research and we are planning to do. Due to time constraints and the complexity of integrating all suggested ESG aspects comprehensively, we have not included some of the detailed suggestions. These limitations have been discussed and acknowledged in the revised limitations section of our manuscript.

Thank you for your valuable feedback, which has significantly enhanced the depth and relevance of our study.

6. PLOS authors have the option to publish the peer review history of their article (what does this mean?). If published, this will include your full peer review and any attached files.

Do you want your identity to be public for this peer review? For information about this choice, including consent withdrawal, please see our Privacy Policy.

Reviewer #1: No

Reviewer #2: Yes: Sergey Barykin We will not be publishing the peer review history of our article. Therefore, the identity of the reviewers will remain anonymous as per the default policy.

Journal Requirements: 

"This study received support from the Research project of the Zhejiang Provincial Department of Education (Y202351899) and Zhejiang Federation of Humanities and Social Sciences Circles Base research projects (2014JDZ01)."

1. We have reviewed and revised our manuscript to ensure it meets PLOS ONE's style requirements, including those for file naming. We have utilized the provided PLOS ONE style templates to format the main body, title, authors, and affiliations sections accordingly.

2. The statement is correct. "The funders had no role in study design, data collection and analysis, decision to publish, or preparation of the manuscript." We will include this statement in the cover letter. Thank you for updating it in the online submission form on our behalf.

3. We have reviewed our reference list to ensure that it is complete and correct. We have verified that none of the cited papers have been retracted.

While revising your submission, please upload your figure files to the Preflight Analysis and Conversion Engine (PACE) digital diagnostic tool, https://pacev2.apexcovantage.com/. PACE helps ensure that figures meet PLOS requirements. To use PACE, you must first register as a user. Registration is free. Then, login and navigate to the UPLOAD tab, where you will find detailed instructions on how to use the tool. If you encounter any issues or have any questions when using PACE, please email PLOS at <a href="mailto:figures@plos.org">figures@plos.org. Please note that Supporting Information files do not need this step.

As requested, we have uploaded our figure files to the Preflight Analysis and Conversion Engine (PACE) digital diagnostic tool. All figures have been checked and meet the PLOS requirements. Also, replace the figures with the converted ones using Conversion Engine (PACE). Thank you for providing this tool, which has helped us ensure the quality and compliance of our figures.

---

## [Decision Letter · Decision Letter 1]

19 Aug 2024

Analysis of the Impact of RCEP on the Industrial and Innovation Chains of China's Textile and Clothing Industry

PONE-D-24-13065R1

Dear Dr. Kumarasinghe,

We’re pleased to inform you that your manuscript has been judged scientifically suitable for publication and will be formally accepted for publication once it meets all outstanding technical requirements.

Kind regards,

Muhammad Hashim, PhD

Academic Editor

PLOS ONE

Additional Editor Comments (optional):

Reviewers' comments:

Reviewer's Responses to Questions

**Comments to the Author**

1. If the authors have adequately addressed your comments raised in a previous round of review and you feel that this manuscript is now acceptable for publication, you may indicate that here to bypass the “Comments to the Author” section, enter your conflict of interest statement in the “Confidential to Editor” section, and submit your "Accept" recommendation.

Reviewer #1: All comments have been addressed

Reviewer #2: All comments have been addressed

2. Is the manuscript technically sound, and do the data support the conclusions?

Reviewer #1: Yes

Reviewer #2: Yes

3. Has the statistical analysis been performed appropriately and rigorously? 

Reviewer #1: Yes

Reviewer #2: Yes

4. Have the authors made all data underlying the findings in their manuscript fully available?

Reviewer #1: Yes

Reviewer #2: Yes

5. Is the manuscript presented in an intelligible fashion and written in standard English?

Reviewer #1: Yes

Reviewer #2: Yes

6. Review Comments to the Author

Reviewer #1: I have studied the revised manuscript and authors responses and suggest acceptance of the article in current form.

Reviewer #2: Dear Author,

Thank you for submitting exciting article. Please consider the sustainability concept regarding the research aim.

7. PLOS authors have the option to publish the peer review history of their article (what does this mean?). If published, this will include your full peer review and any attached files.

Reviewer #1: **Yes: **Asfandyar Khan

Reviewer #2: **Yes: **Sergey Barykin

---

## [Editor Report · Acceptance letter]

23 Aug 2024

PONE-D-24-13065R1 

PLOS ONE

Dear Dr. Kumarasinghe, 

I'm pleased to inform you that your manuscript has been deemed suitable for publication in PLOS ONE. Congratulations! Your manuscript is now being handed over to our production team.

Kind regards, 

on behalf of

Dr. Muhammad Hashim 

Academic Editor

PLOS ONE